# Perceptions of yellow fever emergency mass vaccinations among vulnerable groups in Uganda: A qualitative study

**Lena Huebl**[1,2]*, **Aloysious Nnyombi**[3], **Aban Kihumuro**[4], **Denis Lukwago**[5], **Eddy Walakira**[3], **Ruth Kutalek**[1]

**1** Unit Medical Anthropology and Global Health, Department of Social and Preventive Medicine, Center for Public Health, Medical University of Vienna, Vienna, Austria, **2** Department of Tropical Medicine, Bernhard Nocht Institute for Tropical Medicine & I Department of Medicine, University Medical Center Hamburg-Eppendorf, Hamburg, Germany, **3** Department of Social Work and Social Administration, Makerere University, Kampala, Uganda, **4** Department of Nursing and Health Sciences, Bishop Stuart University, Mbarara, Uganda, **5** Cluster Monitoring and Evaluation Lead, Rakai Health Sciences Program, Masaka, Uganda

* l.huebl@uke.de

**Data Availability Statement:** Relevant data are included in the manuscript as citation excerpts. Full interview transcripts cannot be shared publicly

## Abstract

### Background

Yellow fever (YF), a mosquito-borne viral hemorrhagic fever, is endemic in Uganda and causes frequent outbreaks. A total of 1.6 million people were vaccinated during emergency mass immunization campaigns in 2011 and 2016. This study explored local perceptions of YF emergency mass immunization among vulnerable groups to inform future vaccination campaigns.

### Methodology

In this qualitative study, we conducted 43 semi-structured interviews, 4 focus group discussions, and 10 expert interviews with 76 participants. Data were collected in six affected districts with emergency mass vaccination. We included vulnerable groups (people $\geq$ 65 years and pregnant women) who are typically excluded from YF vaccination except during mass immunization. Data analysis was conducted using grounded theory. Inductive coding was utilized, progressing through open, axial, and selective coding.

### Principal findings

Participants relied on community sources for information about the YF mass vaccination. Information was disseminated door-to-door, in community spaces, during religious gatherings, and on the radio. However, most respondents had no knowledge of the vaccine, and it was unclear to them whether a booster dose was required. In addition, the simultaneous presidential election during the mass vaccination campaign led to suspicion and resistance to vaccination. The lack of reliable and trustworthy information and the politicization of vaccination campaigns reinforced mistrust of YF vaccines.

because of ethical reasons (public availability would compromise study participants' privacy). Data are available from the Ethics Committee of the Medical University of Vienna (Borschkegasse 8b/ E06, 1090 Vienna, Austria, +43 1 40400 21470, ethik-kom@meduniwien.ac.at) or from the Center for Public Health at the Medical University of Vienna (Kinderspitalgasse 15/ 1. Stock, 1090 Vienna, Austria, +431 40160-34881, https://public-health.meduniwien.ac.at/ueber-uns/kontakt/) or from the Bernhard Nocht Institute for Tropical Medicine, Bernhard-Nocht-Straße 74, 20359 Hamburg, Germany, +49 40 4285380-0, bni@bnitm.de) for researchers who meet the criteria for access to confidential data.

**Funding:** The author(s) received no specific funding for this work.

**Competing interests:** The authors have declared that no competing interests exist.

## Conclusions/significance

People in remote areas affected by YF outbreaks rely on community sources of information. We therefore recommend improving health education, communication, and engagement through respected and trusted community members. Vaccination campaigns can never be seen as detached from political systems and power relations.

## Author summary

Yellow fever is transmitted by mosquitoes in tropical regions of South America and Africa. It can cause severe illness and death. Yellow fever can be prevented with a vaccine. However, people in several affected regions are not vaccinated because the vaccine is expensive and not part of routine immunization. For them, access to the vaccine is only possible during emergency mass vaccination campaigns when an outbreak occurs. In this study, we explored local perceptions of emergency yellow fever mass vaccination among vulnerable groups (people over 65 years and pregnant women) in Uganda to understand better how these people have access to vaccine information, what information reaches affected communities, what motivates people to be vaccinated, and what political motives may influence vaccination programs. Despite extensive campaigns, we found that the information reaching communities varied widely, with people relying heavily on community sources. In addition, the lack of trustworthy information and the politicization of vaccination campaigns increased mistrust of yellow fever vaccines. We also found that informed consent is only possible if the principle of vaccination—protection from disease —is understood. Awareness campaigns should focus on a broad understanding of the importance of immunization. Those involved in organizing campaigns should be aware of the potential impact of politicization on vaccine uptake.

## Introduction

Yellow fever (YF), a mosquito-borne viral hemorrhagic fever, is endemic in tropical regions of South America and Africa [1]. YF is preventable with a live attenuated vaccine [2], and proof of vaccination is required for international travel to and from endemic countries [3]. It is estimated that 92.2% of the global disease burden occurs in Africa, partly due to suboptimal vaccine coverage [4]. The YF vaccine is expensive and has not been introduced into routine immunization programs in several endemic countries. As a result, access to the YF vaccine for many people in high-risk areas such as Uganda is currently limited to emergency mass vaccination campaigns [5]. In recent years, YF has re-emerged in West and Central- Africa in areas where large-scale mass vaccination was previously conducted due to immunity gaps caused by population movements and lack of routine immunization programs [6]. In addition, numerous YF outbreaks have been reported since 2016 in non-endemic and endemic areas with historically low YF virus activity [7,8], all with low vaccination coverage [9]. It is estimated that 400–500 million unvaccinated people now live in at-risk areas [7]. In response to re-emerging YF outbreaks, a global strategy to eliminate yellow fever epidemics by 2026 (EYE strategy) was launched in 2017 [5].

Although YF surveillance is routinely performed [10], it has been estimated that most cases of YF are asymptomatic to mild, and only severe cases are detected and reported [11]. Thus, small outbreaks of YF may be missed [12]. In northern Uganda, people were confined to IDP (Internally Displaced People) camps due to two decades of civil war until 2008. During the

2010 outbreak, people had just returned home and needed to clear their land to begin agricultural activities [13]. In response to the 2010 YF outbreak, the largest to date in Uganda, mass vaccination was conducted in January 2011 in the northern districts of Abim, Agogo, Kitgum, Lamwo, and Pader [14]. In 2016, mass vaccination was undertaken in Masaka, Kalangala, and Rukungiri districts in response to smaller YF outbreaks in central and southwestern Uganda [15]. All persons over six months, including pregnant women, were eligible for free YF mass vaccination [14,16]. Overall, 1.6 million people were vaccinated in reactive mass vaccination campaigns in response to outbreaks in 2010 and 2016 [14,16].

Since the outbreaks mentioned above, several YF outbreaks with subsequent mass vaccination have been reported in various districts in Uganda, including Masaka and Koboko in 2019, Buliisa, Maracha, and Moyo in 2020, and Wakiso in 2022 [17]. In response to a surge in reported YF outbreaks in Uganda, the YF vaccine was implemented into the routine immunization program in October 2022 [18], followed by phased mass vaccination campaigns (PMVCs) [17]. The first phase of the preventive mass vaccination campaign for YF was launched on the 8th of June 2023, which aimed to vaccinate 13.3 million Ugandans [19]. A second phase of the preventive mass vaccination campaign against YF will be carried out in April 2024 [20]. Participatory designs that understand local perceptions of immunization and community ownership are essential to ensure equitable access to immunization programs. In addition, local perceptions of vaccine risk concerns must be understood for immunization campaigns to reach their full potential [21].

In this qualitative study, we aimed to explore local perceptions of emergency YF mass immunization in response to two YF outbreaks in Uganda. The study included vulnerable groups, such as pregnant women and the elderly, who are typically excluded from YF vaccination except during mass immunization. The main research questions were to understand how study participants had access to vaccine information and what information reached the affected communities. We also wanted to know how knowledge of the YF vaccine affected motivation to be vaccinated. People better informed about health and prevention issues have advantages in accessing health services, thereby contributing to health equity [22]. In addition, we sought to explore perceptions of vaccines and how they relate to the increasing politicization of immunization campaigns. If the global health community and relevant stakeholders know what perceptions and barriers exist, health programs can be better adapted to reach vulnerable groups. To our knowledge, there has been no study of perceptions of YF vaccination in Uganda.

## Methods

### Ethics statement

Ethical approval was obtained from the National Council for Science and Technology (SS4362), the Mildmay Uganda Research Ethics Committee (#REC REF 0504–2017), and the Medical University of Vienna (EK 1284/2017). The data collected will be kept strictly confidential. All participants were given a unique ID code instead of a name. Before data collection, participants were informed about the project's objectives and the study's purpose. Informed consent was provided in their native language, carefully read to them in their preferred language, and signed or thumb printed. Participation was voluntary, and they were informed that they could withdraw at any time and that their responses would be deleted. All participants permitted their interviews to be audio-recorded.

### Study design

A grounded theory approach inspired by Strauss & Corbin was applied to guide the study design [23]. With its focus on developing new theories to explain social phenomena, grounded

theory is suitable for deepening our understanding of the experience and perceptions of YF mass vaccination campaigns. We conducted a qualitative study between August and December 2017 in two study sites, which had previously conducted reactive emergency mass vaccination for YF outbreaks. Study site 1 included the districts of Kitgum, Lamwo, and Pader, where the largest YF outbreak to date in Uganda occurred in 2010. Study site 2 included the districts of Masaka, Kalangala, and Rukungiri, where smaller YF outbreaks were reported in 2016 (Fig 1. Study sites in Uganda). The study was conducted according to COREQ guidelines to ensure data quality and credibility [24].

We included only vulnerable groups, such as pregnant women and the elderly, who were vaccinated during the YF mass vaccination campaigns. Our aim was to understand their perceptions of YF vaccination better. The focus was not on understanding vaccination refusal but on what information reached this vulnerable group and what motivated them to be vaccinated. Most residents in the affected districts were vaccinated against YF as part of mass vaccination campaigns. They received a YF card to prove their vaccination status and date. At study site 2, where mass vaccination occurred only a year before this study, many pregnant women still possessed YF vaccination cards. The information on the vaccination card made it possible to check when the vaccination took place during pregnancy. With this information and the child's current age, it was possible to check which trimester the YF vaccination was administered. Our study included only vulnerable groups, specifically the elderly and pregnant women. At study site 1 in northern Uganda, only elderly individuals were interviewed. We aimed to compare different YF vaccination campaigns in Uganda, including the largest YF outbreak. The outbreak occurred in a region in the north of the country that had previously been affected by civil war. Even though the outbreak occurred seven years ago and the memories of the older participants may be limited. The focus was not on vaccine refusal but on understanding perceptions of YF vaccination among vulnerable groups, who typically only have access to the YF vaccine during mass vaccination. These findings may optimize future YF mass immunization and vaccination campaigns in vulnerable groups.

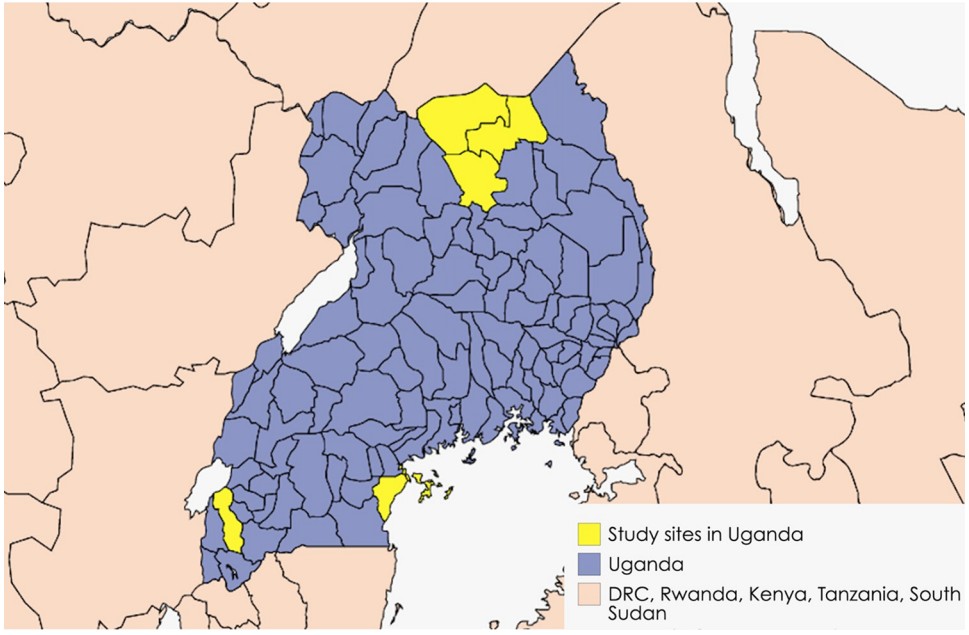

**Fig 1. Study sites in Uganda, created with MapChart, https://www.mapchart.net/.**

## Study setting

Study site 1 in northern Uganda was devastated by two decades of civil war, with people confined to IDP camps until 2008. During the civil war, northern Uganda experienced significant epidemics, including Ebola virus disease (EVD) and hepatitis E. Gulu district experienced the largest EVD outbreak in Uganda between 2000 and 2001, with 425 confirmed cases and 224 deaths [25]. Meanwhile, Kitgum district experienced one of the largest hepatitis E outbreaks in the world between 2007 and 2009, with >10,196 cases of acute jaundice and 160 deaths [26]. At the time of the 2010 YF outbreak, people had just returned to their original homes from IDP camps [13]. In contrast, study site 2 has not been affected by war or significant epidemics. In addition, central and western Uganda are wealthier areas with better health indicators [27] and higher literacy rates [28].

## Participant recruitment

We conducted semi-structured interviews, focus group discussions (FGDs), and expert interviews with 76 participants (Fig 2. Flowchart of study participants). Interview participants had to be older than 18 and were purposively selected from six affected districts. Participants of semi-structured interviews and FGDs had been vaccinated against YF during emergency mass vaccination campaigns. We deliberately included the elderly and pregnant women, usually excluded from YF vaccination. At study site 1, elderly men and women were included. At study site 2, in addition to elderly men and women, we included pregnant women. To verify the trimester of pregnancy at the time of vaccination, we cross-checked the reported gestational age with the date of vaccination on the YF cards and the child's age at the time of the interview.

## Field approach

Data collection took place in the six districts mentioned above. First, permission to conduct research in the district had to be obtained from each district's District Health Officer (DHO). To interview people in rural villages, research was conducted during the rainy season when people were closer to their homes and not scattered working in their fields. Field assistants who were local and familiar with the language, culture, and study site helped with field access and data collection. To reach these villages, we used a car in study site 1 and a motorcycle in study site 2. Especially in northern Uganda, people were scattered and far apart. We drove

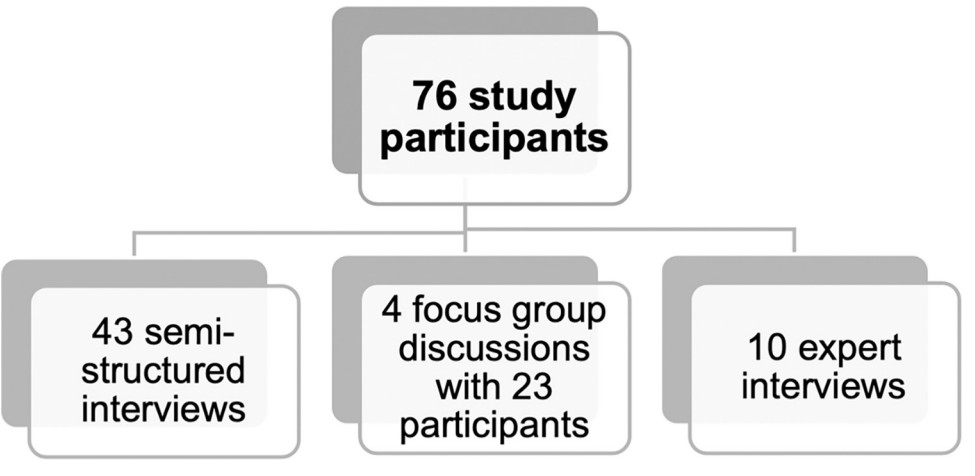

**Fig 2. Flowchart of study participants.**

along main roads in rural areas and interviewed people in villages accessible by road. We also met with local leaders and Village Health Teams (VHTs) who know their communities well. They helped us to reach participants in even more remote locations, accessible by motorcycle or on foot, who met our inclusion criteria. In addition, local leaders and VHTs assisted with FGD recruitment and identification of pregnant women.

## Data collection

We sampled semi-structured interview participants from different religious backgrounds, ages, and marital statuses to capture various perceptions. We also conducted four FGDs with pregnant women, elderly men, and elderly women. FGD participants were required to be homogeneous regarding gender, age, and social status. Overall, FGDs are an important complement to semi-structured interviews. New insights can be generated when participants feel safe to discuss freely among their peers in their native language or are even encouraged by other group members. Interview guides were developed jointly by the study leader (LH) and supervisors (RK, EW). Different interview guides were used for semi-structured interviews, FGDs, and expert interviews.

Experienced interviewers conducted all interviews, the study leader—a postgraduate student in public health- and trained Ugandan field assistants with university degrees in social sciences. Two male and one female field assistant were trained and had previously conducted qualitative interviews. In addition, one male field assistant had experience mapping out and reaching remote villages. All four field assistants were fluent in English. Two were Acholi native speakers, one was a Rukiga native speaker, and one was a Luganda native speaker. All field assistants received additional training from the study leader, including information on informed consent, interview guides, principles of semi-structured interviews, FGDs, and ethical issues.

We interviewed vulnerable groups, including the elderly and pregnant women in rural villages, who don't usually have the opportunity to share their thoughts. Participants were eager to talk and said they felt important because we asked about their beliefs. Nevertheless, hierarchical power relations between researchers and participants must be considered [29]. One older woman terminated the interview early because she felt she had nothing of value to share.

Semi-structured interviews were conducted in or near the respondents' homes. In addition, FGDs were organized by VHTs in health centers or available community spaces. Semi-structured interviews and FGDs were conducted face-to-face in native languages at study site 1 in the Nilotic language Acholi and study site 2 in the Bantu languages of Luganda (Masaka and Kalangala districts) and Rukiga (Rukungiri district). LH conducted all semi-structured interviews with a field assistant who translated into English. FGDs were conducted by field assistants in the native language. After the interview, participants in semi-structured interviews were paid 35,000 Ugandan shillings (UGX), equivalent to USD 10, for their time spent in the interview. Participants in FGDs were compensated with refreshments, UGX 7,000, and travel expenses. Experts received no compensation.

All expert interviews were conducted in the workplace, scheduled in advance, and conducted face-to-face by the study leader in English. Overall, interviews and discussions were non-recurrent and lasted between 30 and 60 minutes. Data were collected until saturation was reached. All interviews and FGDs were audio-recorded, translated, and transcribed verbatim.

The study was conducted with the assistance of four field assistants, each of whom was a native speaker of the respective region. The semi-structured interviews were conducted with a field assistant who provided English translations. LH transcribed the English audio files of semi-structured and expert interviews. In case of ambiguity, the translator was consulted. The

researchers discussed the results with field assistants. The field assistants translated and transcribed the FGDs into English. All transcripts were checked for completeness and imported into Atlas.ti software for qualitative content analysis.

## Data analysis

Data analysis was conducted using the grounded theory described by Strauss & Corbin [30]. Data analysis began in the field by reviewing and refining the interview questionnaires. Field assistants helped with access to the field, data collection, analysis, refining the questions, and overcoming problems at the study sites. Interview results were comprehensively discussed with the field assistants, and their feedback was sought to reduce misunderstandings. New findings were verified in the subsequent interviews and FGDs to confirm the researchers' understanding. Data collection, coding, analysis, and manuscript drafting were carried out by the first author (LH), with support from field assistants. LH conducted coding and data analysis in English in consultation with supervisor RK. Atlas.ti version 8.4.3 software was used for data management [31]. We used inductive coding, and coding progressed through open, axial, and selective coding [23]. Through constant comparison of elements in one data source with elements in another, similarities are identified, and a theory is generated [32]. Thus, codes were clustered into categories and constantly compared. In the process, categories and a core concept developed, and a theory emerged. An example of coding is shown in the results section in Table 1. Findings and new ideas were written down in analytical memos as they emerged. Triangulation was used to compare findings between participants in the FGDs, semi-structured interviews, and expert interviews to identify similarities or differences in their perceptions [33]. Results were shared and discussed comprehensively with the Ugandan research team (AN, AK, DL, and EW). Due to difficulty reaching study participants, they were not included in the analysis, and no member checking was performed. This study was part of a larger research question. Thus, this section includes topics on sources of information, knowledge of the YF vaccine, perceptions of mass immunization, and motivation for vaccination. Quotes have been edited to improve readability while maintaining their structure and linguistic intent.

## Robustness and reflectivity

We used triangulation to increase the robustness of our findings [34]. We used semi-structured interviews, FGDs, and key informant interviews for methodological triangulation. We also used data triangulation: collecting data from different populations, affected regions, and YF outbreaks. Data from semi-structured interviews, key informants, and FGDs were analyzed separately. Findings were consistent across data sources and methods. Nevertheless, it is important to note that the vaccination campaign occurred seven years ago in northern Uganda. As a result, some study participants may not accurately recall the events.

Awareness of positionality can bring latent power dynamics to the surface [35]; the study leader, a young female medical doctor from Europe, was an outsider. Having spent part of her medical training in the Global South, she was familiar with inequalities in access to health care. In particular, how privileged Western countries have easy access to the YF vaccine for travel while people living in endemic areas are not protected. In addition, she was familiar with study site 1 from a field study in northern Uganda five years earlier. She lived with Ugandans in the study areas, and field notes were used for reflection. Nevertheless, there were power dynamics to consider:

The study leader grew up in a predominantly white environment in Europe; as a European researcher conducting a study with Ugandan participants, power dynamics based on race may have affected the openness of discussion and comfort level. To reduce racial and class bias,

**Table 1. Overview of grounded theory coding analysis with main categories.**

| Open coding (only most important codes) | Axial coding | Selective coding | Core category |
|---|---|---|---|
| Medical expertise<br>Sources for medical experts<br>Campaign strategies<br>Target groups<br>Vaccine dosing<br>Troubleshooting<br>Booster doses in vulnerable groups<br>Global vaccine shortage<br>Outbreaks in other districts and countries | Knowledge gap among medical experts | **Unequal access to information** | **Knowledge hierarchy** |
| Information distributed through awareness campaigns<br>Access to health education<br>Access to radio programs<br>Information from friends and family<br>Community sources of information<br>Usefulness of information distributed | Multiple sources of information | | |
| No understanding<br>Modern medicine<br>To prevent getting YF in the future<br>Treatment of YF | Purpose of a vaccine | **Knowledge gap on YF vaccination** | |
| Requirement for booster doses<br>Vulnerable groups<br>Trust in medical experts<br>Longevity of protection | Vaccine dosing | | |
| Protection of deadly disease<br>Anxiety about future health<br>Normally no access to the YF vaccine<br>Vaccine costs<br>Everyone else was vaccinated | Risk perception | **Motivation for vaccination** | |
| Travel requirements<br>Access to health facilities and schools<br>Inconsistent recommendations | Proof of vaccination | | |
| No Access to education<br>Pressure to conform<br>Top-down communication<br>Lack of individual responsibility<br>Trust in HCWs and medical experts<br>Dependency on medical, political, and religious leaders<br>Medical experts from central Uganda | Reliance on health experts | **Power relations between medical experts and the public** | |
| Vaccine rumors<br>Presidential election<br>Color association of certificates with the ruling party<br>Opposing party member<br>Proximity to other immunization campaigns<br>Implementation of new target groups<br>Power dynamics between government and marginalized regions | Distrust in the government | **Politicization of campaigns** | |

field assistants from the region who spoke the language and were familiar with the culture and customs translated during the interview. This helped us understand the topic and reduce misunderstandings during the interview. As a young physician who has experienced the loss of loved ones and patients, this experience may have led to an open conversation about sudden death from YF and fear of pregnancy complications. In addition, being an outsider to the field and not involved in politics may have opened the discussion about the impact of politics on YF vaccination campaigns.

However, power dynamics were apparent due to asymmetry and hierarchy [36]. This power dynamic was particularly visible between the medical experts, all men, and the young

female researcher. In addition, there was a power dynamic within the research team, as the researchers from the Medical University of Vienna were on the outside and relied on the Ugandan team on the inside to successfully conduct this field study. However, it is important to consider the positionality of our co-researchers in Uganda. Specifically, we must address gender issues. Male field assistants were responsible for translating interviews with pregnant women and conducting FGDs with them. This may have led to reluctance on the part of the women to disclose certain aspects of their experiences and beliefs about such a sensitive topic in the presence of men. The field assistants who aided us in our research were fluent in the local native language and were often from the same district as our participants. This allowed them to have a deep understanding of the local customs and challenges of the area. However, it is possible that their proximity to the district may have limited their ability to view certain aspects from a broader perspective. The ethnic differences among our Ugandan co-researchers must also be taken into consideration. In addition, we experienced a hierarchy of knowledge, as medical experts had different expertise than the general population. There was also a power dynamic between northern and central Uganda, as medical experts in northern Uganda are often not from the region but from the south and belong to a different tribe.

## Results

Five main categories emerged from the analysis (Table 1. Overview of coding analysis with main categories): unequal access to information, knowledge gap on YF vaccination, different motivations for vaccination, power relations between medical experts and the general population, and politicization of vaccination campaigns. Our findings are illustrated with quotes from participants.

### Characteristics of study participants

Seventy-six participants participated in this study (Table 2–4. Characteristics of participants). Of these, 29 participants were from northern Uganda, 27 from central Uganda, and 20 from southwestern Uganda. Sixty-three percent of all participants were female. Twenty women were pregnant at the time of receiving the YF vaccine. In northern Uganda, participants belonged predominantly to Nilotic language groups and Bantu groups in central- and southwestern Uganda. Ninety percent of the experts were male and had more than five years of work experience.

### Unequal access to health information

**Knowledge gap among health experts.** YF is one of the diseases tracked in weekly epidemiological surveillance by district health offices, and both suspected and confirmed cases are reported to the Ministry of Health. However, health experts explained that they had little knowledge of YF because it was rarely seen in Uganda before the two outbreaks. Some of the medical experts interviewed believed that YF had been eradicated. Furthermore, medical experts expressed that most had no first-hand experience with YF before the outbreak and relied on medical books and the Ministry of Health for information.

> We were surprised because YF had been eradicated a long time ago. And we were trying to see how it was transmitted. We had to educate ourselves.—male health expert with fifteen years of work experience, Lamwo

Similarly, at the community level, Health Care Workers (HCWs), VHTs, and Local Chairmen (LCs) were often unaware of YF before the outbreaks, as a health expert explained:

**Table 2. Socio-demographic characteristics of semi-structured interview participants.**

| Semi-structured interviews (n = 43) | |  |
|---|---|---|
| **Sex:** | 33% male, 67% female | |
| **Age:** | median age 70 years (range 21–88 years) | |
| **Pregnant women:** | n = 13 (District Masaka, Kalangala and Rukungiri) | |
| **Marital status:** | Married: n = 33<br>Widowed: n = 7<br>Single/divorced/no data: n = 3 | |
| **Children:** | median 6 children (range 0–30) | |
| **Education:** | Illiterate: n = 10<br>Primary: n = 17<br>Secondary: n = 12<br>College degree: n = 4 | |
| **Occupation:** | Farmer: n = 25<br>Retired: n = 6<br>Housewife: n = 5<br>Business: n = 3<br>Teacher: n = 2<br>Other: n = 2 | |
| **Region:** | **Northern Uganda (n = 15)** | |
| | *District* | *Sub county* |
| | **Pader** | Latanya, Pader Town Council, Ogom |
| | **Kitgum** | Labongo Amida, Omiya Anyima, Lagoro |
| | **Lamwo** | Padibe Town Council, Padibe West, Paloga |
| | **Central Uganda (n = 18)** | |
| | *District* | *Sub county* |
| | **Masaka** | Kyanamukaaka, Kyesiga, Masaka municipality, Buwunga, Mukungwe |
| | **Kalangala** | Bujjumba, Mugoye |
| | **Southwestern Uganda (n = 10)** | |
| | *District* | *Sub county* |
| | **Rukungiri** | Kebisoni, Nyakagyeme, Bwambara, Bugangari, Nyarushanje, Buyanja, Buhunga |

**Table 3. Participants of focus group discussions.**

| Focus group discussion (n = 23) | | | |
|---|---|---|---|
| **Participants** | | **Age (median)** | **District** |
| **Elderly men** | n = 5 | 70 years (range 62–85) | Kitgum |
| **Elderly women** | n = 5 | 62 years (range 60–69) | Lamwo |
| | n = 6 | 54 years (range 50–65) | Masaka |
| **Pregnant women** | n = 7 | 30.5 years (range 25–34) | Rukungiri |

**Table 4. Participants of expert interviews.**

| Expert interviews (n = 10) | |
|---|---|
| **Sex:** | 90% male |
| **Occupation:** | DHO, Assistant DHO, Surveillance focal person, Health worker, Rapid response team, Medical Officer |
| **Years of experience:** | < 5 years: n = 2<br>6–10 years: n = 3<br>11–15 years: n = 2<br>16–20 years: n = 3 |

*When we talked about it on the radio, it was a strange disease. Many people have not heard of yellow fever. There are some even these old people who would say 'Ah those are the disease we would hear in other countries like forty years ago'. So, it was seen as a new disease, although it has been one of the diseases that we track on our weekly epidemiology surveillance. The list we have of the diseases that are epidemically prone, which are supposed to be monitored on a weekly basis. Eh, but the community level was not aware and until we did the sensitization VHTs had no knowledge about YF.–a male medical expert with eleven years of work experience, Rukungiri*

Health experts explained that HCWs in their affected districts were trained in a two-day workshop on the signs and symptoms of YF and how to administer the vaccine. In addition, a modified case definition was distributed to health centers in local languages and English. A medical expert explained the case definition at the community level:

*The case definition was that any person presenting with jaundice and abdominal pain should be reported to the nearest health facilities as a community case definition.–a male medical expert with seven years of work experience, Masaka*

Furthermore, health experts at study site 2 elaborated that as part of local health education, VHTs and LCs were informed about the signs and symptoms of YF and given a simple case definition to help them identify suspected cases in their communities and report them to HCWs. Community surveillance by VHTs and HCWs could then report suspected cases to a mobile hotline, which provided direct contact with medical experts at the district health office. In addition, health experts stated that they often used private resources such as cars, fuel, and mobile phone credit to visit suspected cases.

**Information distributed through awareness campaigns.**    Medical experts explained that they relied heavily on LCs and VHTs to raise awareness and mobilize for mass vaccination, especially in hard-to-reach areas and villages. In communities with cases of YF, local authorities also conducted door-to-door health education. As explained by a woman whose family member died from YF during the outbreak:

*Our family was sensitized about YF and its vaccine by district health personnel. Afterward, we accepted to get vaccinated for YF. After receiving information about YF we couldn't go on believing it was due to witchcraft. However, many people could say it (YF) is witchcraft and they had to seek other treatments. Although there are better methods, actually a vaccination, that protects people from the disease (YF).– 65-year-old female farmer, Masaka*

In addition, posters were displayed at health facilities and vaccination sites in the local language, Acholi, at study site 1 and in English at study site 2. These posters showed how to prevent mosquito bites, how to destroy mosquito breeding sites, how to seek early treatment, and where to get vaccinated against YF.

As several participants explained, they had been informed about the signs of YF and the upcoming mass vaccination as part of the awareness campaigns. Furthermore, they elaborated that they were also advised to seek medical care if sick, to clean their compound, to use latrines, and to consume boiled water and well-cooked food, which are not specific preventive measures against YF.

**Access to health education programs.**    At study site 1, participants explained that during the 2010 YF outbreak, many of them lived in IDP camps in northern Uganda. They elaborated

on how HCWs used a door-to-door approach to educate IDP camp residents about the outbreak, disease prevention, and the upcoming mass vaccination campaign. In a group discussion, elderly women discussed how they had been sensitized:

*They first called people and sensitized us. We were sensitized from here Padibe health center IV, they told us that if you contract YF and at the same time happen to be pregnant, it can easily kill you. When the issue of YF came in, they started showing a YF film for us to understand how dangerous YF is on human beings, as a mean of passing out information about YF. If you watch those films, you will just go by yourself to the hospital to get the vaccination. Our Local chairman came to where we were digging and told us that we should go and get vaccinated because they are vaccinating everyone for yellow fever.–FGDs with elderly women, Lamwo*

However, when asked in a group discussion, an elderly man explained that although they were mobilized to be vaccinated, they did not receive sufficient information about YF.

*When we were being vaccinated, there wasn't adequate information about yellow fever. They did not give us any information apart from just the call to people to come and get vaccinated. You know the medical people; the doctors should have at least told us. (. . .) And what causes it and how it is transmitted. (. . .) We know of other diseases like HIV/AIDS, polio, malaria, and others that are taught at school, but yellow fever is not.–FGDs with elderly men, Kitgum*

In contrast, during the 2016 YF outbreak, people had broader access to health education programs, as radio access was widespread in study site 2. This was repeatedly observed during our visits to respondents' homes. In addition, several participants described how they were sensitized and mobilized through health education programs and awareness campaigns on radio talk shows.

*They put announcements and radio talk shows to educate, to tell people about yellow fever and they also used health workers at the health facilities to move around villages and communities telling people about yellow fever. -34-year-old businesswoman, vaccinated during pregnancy, Rukungiri*

A health expert explained that at study site 2, information was also disseminated through megaphones provided by local councils as part of the outreach health education program at the regional level. Participants who had access to these health education programs widely agreed that information about the outbreak, the affected areas, the occurrence of deaths, and when, where, and for whom mass vaccination would occur was disseminated. In addition, they stated that they were informed about how YF is transmitted and were advised to sleep under mosquito nets.

*We heard it over the radio and also that in the neighboring subcounty in Kebisoni it (YF) killed two people, who died at the health facility and people would come back to the community and talk. And also, again that people went for vaccination fast. When we reached the vaccination sites there were posters so that they would read and come to tell them that yellow fever exists, and it kills.– 21-year-old female farmer, vaccinated during pregnancy, Rukungiri*

However, participants, especially those in remote villages, explained that even after the awareness campaigns, they still felt that they needed to be adequately informed about the cause of the outbreak and the transmission of YF.

*So, the information was lacking, simply because they were not informed about the cause of the outbreak and what they could do so they don't fall victim of yellow fever during the outbreak.– 27-year-old housewife, vaccinated during pregnancy, Kalangala*

**Reliance on community sources for information.** During our visits to participants' homes, we observed that many did not have access to electricity, particularly in remote villages. Without access to mass media, people relied on community sources for information. Medical experts in northern Uganda explained that they used a one-way communication strategy. They elaborated that HCWs instructed LCs to inform their communities about the YF outbreak and reactive mass vaccination. With limited radio access at the time, villagers who did not live in IDP camps and therefore could not be reached by HCWs relied on their LCs and fellow community members for information. An LC from the Pader district explained:

*We were told that yellow fever is coming, but the information came from the government telling us that yellow fever is coming, and everybody should get vaccinated, and we all got vaccinated.– 73-year-old male LC, Pader*

At study site 2, participants explained that health education programs were conducted through radio broadcasts, and information was disseminated through LCs and VHTs. In addition, information about the YF outbreak and the upcoming mass vaccination was distributed through community meetings, churches, mosques, other religious gatherings, and communal areas.

*We were informed via radio and portable public communication that is always used at local levels. - 22-year-old woman vaccinated during pregnancy, Kalangala*

However, in all affected districts, particularly participants who lived in remote villages explained that they did not have access to health education programs and relied heavily on information from fellow community members. Farmers cultivating their fields during mass vaccination explained that they were informed by fellow community members returning from vaccination sites. An elderly farmer described how he arrived late at the vaccination site and missed the sensitization. As a result, he knew nothing about the vaccine or the outbreak except that he had been vaccinated.

*I never got the information, because by the time I reached the health center they had already taught people. I just got the vaccine and went back home. I don't remember any mobilization about yellow fever, but the day I got information about yellow fever was the day they were vaccinated, and I was just called from the garden. Someone told me vaccination is going on, I should come to be vaccinated so I came. - 70-year-old female farmer, Kitgum*

## Knowledge gap on YF vaccination

Despite all the YF campaigns described above, there seemed to be very little knowledge and much confusion about the YF vaccine. Participants repeatedly stated that they had little knowledge of the YF vaccine they received during the mass vaccination.

*That no, I don't know anything about the YF vaccine. Because even when we were vaccinated, we were not told anything.– 68-year-old female farmer, Pader*

Another farmer added that it is modern medicine.

*No, nothing. I know the YF vaccine is modern medicine and that's all.– 70-year-old female farmer, Kitgum*

In addition, several participants explained that they had never heard of the vaccine before the mass immunization.

*I don't know much about the YF vaccination. All I know is that once you are vaccinated it means you have acquired that immunity so that YF will never come to your body. I had not known about the YF vaccine before.– 72-year-old retired male officer, Rukungiri*

Furthermore, it was unclear to several participants whether the vaccine would prevent YF or treat the disease.

*All I know about the vaccine is it is healthy and once given to you (the person with the vaccine), you are free from yellow fever.– 26-year-old housewife, vaccinated during pregnancy, Kalangala*

In northern Uganda, one LC elaborated that they were told that the vaccine would treat YF.

*They told us this vaccine treats yellow fever. That's what we were told. -73-year-old male Local Chairman, Pader*

When asked in interviews, medical experts lacked information about the YF vaccine. One expert narrated that they had been trained to administer the vaccine and that it would provide lifelong protection. However, this was not widely known among the interviewed medical experts. Furthermore, one expert explained that they were not given information on whether a standard or fractional dose (1/5 of the standard dose) was used during emergency vaccination campaigns.

*No, that 1/5th of the dose was administered, we didn't hear about it and then again for us as a district, we are implementors of a government policy. Ministry came and trained us, they said we are going to administer this dose it will give lifelong protection and that was all. So, because we didn't know the dose, about the dosages, about the pharmacology and all that.–a male medical expert with ten years of work experience, Rukungiri*

All medical experts stated they were unaware that fractional doses had been used during similar outbreaks in Angola and the Democratic Republic of the Congo to address the global vaccine shortage in 2016. As repeatedly confirmed, the vaccine dose and the requirements for a booster dose were unclear. At study site 1, several participants explained that they were told they would need a booster dose after ten years, which was consistent with the WHO YF vaccine recommendation at that time.

*During those days, we were only given one (vaccine dose). Just once. Yeah, we were given vaccination just once. After ten years, according to what they (HCW) told us. After ten years then another vaccination may occur. That's how.– 72-year-old retired male police officer, Pader*

However, at study site 2, there was no clear guideline for the interval between the booster doses. Furthermore, whether or not a YF booster dose was needed and at what interval was unknown to most participants. Throughout the interviews and group discussions, perceptions

varied from a single dose to a booster dose after one, three, five, or ten years. An elderly woman explained that she thought it should be three doses with an interval, like the hepatitis vaccine, to get adequate protection because she had never seen a vaccine given in just one dose.

*Normally, with the vaccines, we need to do it three times. I think it should also be three times (for YF). I have never seen any vaccine given for once and all. Never, I see vaccination in children (given one time) with measles, polio, and even now with hepatitis. They do it three times. I also think somebody has to be vaccinated three times and intervals.– 65-year-old female farmer, Lamwo*

Participants widely agreed that the YF vaccination card was an essential document, proof of vaccination that they would need to show at health facilities when sick and for travel purposes when crossing country borders. However, several participants explained that YF cards were necessary to cross district borders and access schools and health facilities. In addition, as one HCW explained, they reinforced the belief that people could not even cross district borders without vaccination.

*Without the vaccine, people will be refused to leave their district. People feared and went for the vaccine.–female Health Care Worker with two years of work experience, Rukungiri*

## Motivation for YF vaccination

Participants and medical experts of all affected districts stated that the massive mobilization and high turnout at YF vaccination sites was unlike any other campaign they had seen in their district. They expressed that this was because trusted community members had sensitized and mobilized people. In addition, vaccinations were conducted over several days at the parish level. According to medical experts, vaccination sites were shared by 3–4 villages, and communities were familiar with these sites (e.g., parish centers, churches, schools, commercial centers, and health centers) as they were typically used for other immunization programs.

*We used to have campaigns of measles, and vaccination of children and then we had designated specific places at churches and schools in every parish. We used the same places because the mothers and the other caregivers were used to those places.–male medical expert, with ten years of work experience, Rukungiri*

According to several medical experts, mass vaccination was voluntary, and anyone interested could be vaccinated. A medical expert at study site 2 explained that people were required to show their national identity cards at the vaccination sites. However, there was no verification of whether they lived in the district and were eligible for vaccination. He explained that as a result, residents from unaffected districts rushed to the mass vaccination sites to be immunized against YF.

*The campaign was based on a voluntary approach. We used statistical data from the Uganda Bureau of Statistics to benchmark our targets. Where any individual could come and seek the vaccination, anyone who had an interest would come. So, we had no way to determine whether, within our own community, some inhabitants were never vaccinated, because these people (from outside the district) filled up the gap; to even surpass the target population.–a male medical expert with nine years of work experience, Masaka*

The central message of the mass vaccination campaign was to protect people from a deadly disease. Overall, the fear of dying from YF was widespread, and several participants from study

site 1 recalled that cases of YF had been reported in the IDP camps where they lived. Several participants had been directly affected by surviving YF and family members dying from YF. A retired officer narrated:

*Four of my neighbors got YF. I accepted because I thought I could get infected with yellow fever. I was scared. -67-year-old retired officer, Lamwo*

In addition, HCWs reinforced this fear, with one HCW explaining that they told people that if they did not get vaccinated, they would die, and all the residents of the district could die.

*We (HCW) told people at vaccination posts that people are dying of YF. And if we don't get that vaccine, people will continue dying and the whole district may die.—female Health Care Worker with two years of work experience, Rukungiri*

Furthermore, the YF vaccine was sought because it was perceived as expensive and had limited access, as it was not part of routine immunization and was required for international travel. A young woman vaccinated against YF during pregnancy explained that she only has access to this vaccine when it is distributed through mass immunization programs.

*Mass vaccination is very good because if they tell us, individuals, we need to get vaccinated on our own you may not be able to afford or access the area where they are vaccinating from. When it is during mass vaccination, they bring the vaccine to the local community. It helps to easily access the vaccination.– 34-year-old female businesswoman, vaccinated during pregnancy, Rukungiri*

In particular, at study site 2, participants stated that obtaining a YF vaccination certificate for travel was one of the dominant incentives for vaccination, in addition to protection from a deadly disease.

*I wanted the yellow card. Simply because you can never cross the borders without a yellow card.– 81-year-old male farmer, Kalangala*

Although most participants did not have the means to travel abroad, the fear of being unable to cross borders in the future motivated them to be vaccinated.

*They told us that once you don't possess that card you can never travel to any other country. You just got to have that, so that you can travel. So, I wasn't certain whether that was true because I have never tried traveling since.– 28-year-old housewife, vaccinated during pregnancy, Masaka*

Thus, one medical expert explained that people from districts not affected by the outbreak rushed to get vaccinated, motivated primarily by a free travel vaccine.

*Interestingly some people were coming from outside the district to get the vaccination. Especially those who were from Kampala and even the neighboring districts to get the vaccine because they knew it is a vaccination, which is also expensive, especially for those who travel outside the country.–a male medical expert with more than ten years of work experience, Kalangala*

Participants widely noted that everyone in their family had been vaccinated against YF during mass vaccination campaigns. Furthermore, several parents believed that YF vaccination was required for their children to attend school. An elderly farmer explained that he had asked his great-grandchildren to return from school to ensure they were vaccinated against YF.

*All the members in the family were vaccinated even those that were in school were told to come back and get vaccinated so that they can return to school. - 81-year-old male farmer, Kalangala*

Children in Uganda are often vaccinated at school as part of the childhood immunization program. Several parents didn't know if their children had been vaccinated against YF at school. However, the parents explained that they did not ask their children or check their YF certificates.

*I had to comply and take my wife and one boy and also two girls in my family for vaccination. The rest (of the children) were in school. I expect that since it was an outbreak the health experts should have moved into schools and vaccinated them (the children) from there. - 76-year-old retired male teacher, Rukungiri*

Other interviews supported this notion.

*I was the only one in the family who went for vaccination. It was schooling time and the kids were in the boarding section. I am not well informed whether my children did receive the vaccine or not. - 84-year-old male farmer, Kalangala*

*I was vaccinated alone because by then I was alone. The children I take care of were all at school. I don't know if the children were vaccinated at school. I didn't ask them. - 75-year-old female farmer, Lamwo*

When asked why family and other community members refused to vaccinate against YF, participants explained that they believed some feared the painful injections and common side effects of vaccination, such as local swelling at the injection site, fever, and myalgia. It was also believed that being out of town, busy at work, and tired of waiting in line were common reasons for missing the vaccination. Particularly in rural areas, several participants explained that the distance to vaccination sites during mass vaccination was too far for them and that they would prefer a door-to-door approach, as expressed by a woman in study site 2.

*More so if it is done at an outreach level some people are not able to access the vaccine simply because some people are kind of a distance away from the nearby health facility, so they are always reluctant to move such long distances to see the health facilities for vaccination. So, doing an outreach or making it door-to-door is kind of good.– 32-year-old female trader, vaccinated during pregnancy, Masaka*

Some participants elaborated that they thought there were different ideologies among community members who opposed the YF vaccine; for example, vaccination was unnecessary for healthy individuals and that anyone who had previously survived YF was protected and did not need additional vaccination. However, people must be educated about YF to make an informed decision.

In addition, several participants explained that it was widely believed that the vaccine could be toxic and harmful, killing weak people, which led to the rejection of the YF vaccine. A woman shared her thoughts on vaccine refusal:

*Some people were just in opposition to vaccination. They refused to attend the vaccination process. Claiming that there are some toxic materials in the vaccination. So, at some point, people might even die after being vaccinated. -31-year-old female farmer, vaccinated during pregnancy, Masaka*

Participants explained that although diseases such as polio and measles are now rare in their communities because of mass immunization of children, this has not affected subsequent mass immunization campaigns. A farmer explained that people want certain medicines and vaccines while others are rejected.

*I don't know why a person wouldn't come back for vaccination, because sometimes they can send for taking (the vaccine) and people don't want it. They want certain medicines and don't want others.– 65-year-old female farmer, Lamwo*

## Power relations between health experts and the general population

Health experts explained that they used a one-way communication strategy to disseminate information. When asked in interviews, there appeared to be top-down communication from HCWs. At study site 1, participants explained that they were predominantly vaccinated because they were mobilized in their communities and told to get vaccinated against YF, a deadly disease.

*It is because the health workers said people should get vaccinated and that's why I got vaccinated. -74-year-old male farmer, Kitgum*

In addition, several participants reported that vaccination was perceived as mandatory, with HCWs telling them that they would die of YF if they were not vaccinated.

*I got vaccinated because HCW told us if you don't get vaccinated you will die. -65-year-old female farmer, Lamwo*

In addition, villagers claimed that they could not refuse medical advice from more educated HCWs.

*You know we are uneducated people we can't reject anything medically, or if we are informed to do so, like vaccination, we must follow the caregivers.– 67-year-old male farmer, Lamwo*

Participants also showed great trust in HCWs and health professionals. For example, participants in northern Uganda were not concerned about inadequate protection when they did not receive their booster dose at the perceived interval.

*They vaccinated them just once that time. Maybe in case they still need to vaccinate. If they come and tell people to go vaccinate again then I would.– 76-year-old female farmer, Pader*

Participants claimed that it was the responsibility of the government and medical experts to know if a booster dose was needed.

*I don't know. It is upon the health workers to decide how they do it. Whether they vaccinate or not. - 72-year-old female, Kitgum*

Furthermore, they lacked individual responsibility and thought that medical experts and HCWs would tell them to vaccinate if necessary.

*I don't know anything about how often and frequently you have to go for the YF vaccination, but I think the medical experts know it better. -84-year-old female farmer, Kalangala*

However, medical experts were unaware that a booster dose was needed for children vaccinated before age two and for women vaccinated during pregnancy. There was also no strategy for follow-up and booster doses.

Furthermore, farmers in remote areas narrated that they were unaware of the extent of the emergency mass vaccination campaigns or the outbreak. Nevertheless, they felt that everyone was expected to be vaccinated. A young woman who was vaccinated during pregnancy explained that she felt forced to be vaccinated because HCWs told her that if she contracted YF, she would die.

*I was forced to take the vaccination because they (HCW) had told us that this disease has come to the area and that it kills people and that once you get it you must die. - 21-year-old farmer, vaccinated during pregnancy, Rukungiri*

Several participants elaborated that they had been vaccinated because everyone else in their community had been vaccinated and that they would not defy a government campaign on their own.

*Everyone got vaccinated and I cannot be the only one, who would defy the government's orders. - 73-year-old female farmer, Rukungiri*

In addition, a participant in western Uganda explained that because everyone else was vaccinated, he feared that if he disobeyed, he would be the only one in his community to become infected and subsequently die from YF.

*It was a government program and what if I would refuse to get vaccinated then I get sick alone? -77-year-old retired male officer, Rukungiri*

### Vaccine Rumors & Politicization

Despite the widespread motivation to vaccinate against YF, distrust of the government led participants to fear that the vaccine could be toxic and harmful. Moreover, some villagers explained that they believed the vaccine could paralyze people and shorten their lives.

*So, someone might not get vaccinated because of their ideology. Some, have an ideology that when you are vaccinated your lifespan is always reduced. They think that through vaccination they are some toxic processes that can be passed on. - 81-year-old male farmer, Masaka*

It was believed that if you were already weak, you could quickly die from the YF vaccine, as a farmer in northern Uganda explained:

*I was hearing some rumors that if you are a weak person and when they vaccinate you with the YF vaccine, you can easily die. -74-year-old male farmer, Kitgum*

In addition, rumors that the YF vaccine could cause miscarriage in pregnant women were widespread among participants. Women at study site 2, who were vaccinated during pregnancy reported miscarriage as their primary concern. However, medical experts expressed that they faced another major challenge: the politicization of YF vaccination. 2016, the emergency mass vaccination coincided with President Museveni's presidential election campaign. Some of the participants explained that rumors had spread that people would be injected with the YF virus at vaccination sites and then die of the disease. They believed that as a government campaign, they could easily reach and kill members of the opposing party by injecting them with the virus. A young woman explained that people refused to be vaccinated because they were afraid of being harmed by the government.

*Another section of the population, they thought that it was Museveni's' (the President of Uganda) plan to harm communities through a governmental vaccination program. Due to the fear of getting harmed by the government on purpose, people did not go for YF vaccination. - 21-year-old female farmer, vaccinated during pregnancy, Rukungiri*

Thus, political motives, such as opposition to the ruling party, led community members to oppose the emergency YF mass vaccination openly. An elderly farmer elaborated that it was all about politics.

*Some people disapproved of the vaccination process because they hate the president who is in power. Simply because whatever the president does, they are just in objection to what his government is thinking of. So, they failed to turn up (for YF vaccination) and they were really defiant.– 81-year-old male farmer, Masaka*

Furthermore, several participants explained that people also questioned why the YF vaccination cards were yellow, the color of Museveni's political party. According to them, people who refused to be associated with the ruling political party rejected the YF vaccination because of the yellow card issued, as explained by a teacher in western Uganda:

*Some people during the election period, who were ignorant, thought that the opposition would think that Museveni, the current president, had brought some medicine to kill them so that they won't be able to vote for their other presidential candidate. It is true that some people thought 'Why are they specifically using the color yellow which is the color of the current president'. Why couldn't they use any other color (for the YF certificates)? People rejected the yellow cards and I think they did not get vaccination because they were given the yellow card before vaccination. So, if someone rejected the yellow card means he wouldn't get vaccinated. He would just go back home. -76-year-old retired male teacher, Rukungiri*

A medical expert clarified that they had to educate people to accept the yellow cards.

*They were saying 'Why did they bring a yellow card? Why not blue or any other color'. And then we explain, that for yellow fever we are using yellow cards. Just for that reason. So, eventually, they accepted, but those were just political opponents.–a male medical expert with ten years of work experience, Rukungiri*

Furthermore, medical experts from study site 2 explained that implementing emergency mass YF vaccination in communities opposed to the ruling party was challenging. They elaborate on how district health workers used a variety of strategies to dispel doubts and encourage

people to be vaccinated against YF. In addition, health experts who had used radio to inform communities about the ongoing YF outbreak invited local leaders and opposition politicians to speak on radio programs. A medical expert explained that people followed suit when community members saw respected political and religious leaders advocating for YF vaccination.

*On day one (of YF vaccination) it was politicized, why Rukungiri? Rukungiri supports the FDC (Forum for Democratic Change, the opposition party), and we had just concluded the elections. (. . .) People were resisting yellow cards, because yellow is the color of the NRM (National Resistance Movement). I had to go to the radio with the Resident District Commissioners and others, there is a mayor here in the district, he is a person of FDC, and he is a very big one. We had to go to him, we had to talk on the radio, and then people would call me even when we had finished the program. (. . .) Then people showed up for vaccination.–a male medical expert with eleven years of work experience, Rukungiri*

Furthermore, a female HCW explained that they sensitized community members at the vaccination sites, and to build confidence in the vaccine's safety, the HCWs were vaccinated in front of the queues.

*People were fearing the vaccine. They thought that since there was yellow fever, then maybe the vaccine was deadly. There were politics they feared. At first, we injected the health workers (with the vaccine) when they were there watching. Then after they were given the vaccine.— female Health Care Worker with two years of work experience, Rukungiri*

Despite being sensitized by trusted community members and having free access to the expensive YF vaccine, one mother of five was initially suspicious about why the community was being mobilized for another mass vaccination when they had just completed a mass polio immunization in the previous months. The young mother explained that rumors were spreading that the vaccine could be toxic and that they were being targeted to be killed by the vaccine.

*In the community, they claimed that those toxic substances within the vaccination are just aimed at ceasing the generation. To kill everyone. So, those were the rumors spreading around and some months back there was a mass vaccination for polio for those aged 5 years and below. So, having brought back the vaccination they thought like. . .No, these people might be against us, to have some plan of having us killed.– 31-year-old female farmer, vaccinated during pregnancy, Masaka*

In addition, adults were targeted for mass immunization for the first time, especially in study site 2. These communities were only familiar with the mass vaccination of children. Therefore, targeting adults for vaccination was a novelty and was perceived as unusual.

*Initially, there were mass vaccination campaigns for infants below five years, but for the elderly, the grown-up there has no mass vaccination been realized before.– 31-year-old female farmer, vaccinated during pregnancy, Masaka*

A mother from a remote village in the Ssese Islands said they had often conducted mass drug administration to treat schistosomiasis, but this was their first mass vaccination of adults. Despite extensive awareness campaigns in the district, the information reached them the day before the mass vaccination. She explained that they felt that HCWs rushed into their

community and vaccinated them without adequate sensitization about the disease and the purpose of the vaccination.

> *My opinion about mass vaccination is positive, but there should be some sensitization and emanation of information about what disease they are going to immunize and why are they immunizing. Simply because at some point the HCW came abruptly and told us tomorrow we are going to vaccinate, and we have to turn up. Not even knowing what the cause was to vaccinate us and things like that.– 27-year-old housewife, vaccinated during pregnancy, Kalangala*

## Discussion

Our results showed that despite extensive YF mass vaccination campaigns, the information reaching affected communities was inconsistent and often fragmented. This information gap ranged from community members not knowing what a vaccine does to HCWs not knowing what they were administering, whether a standard or a fractional YF vaccine had been administered, who should receive a booster dose, and after what interval. Furthermore, our results showed that YF vaccination was motivated by receiving an important vaccination for travel purposes. On the other hand, there was a misconception or misinformation that a YF certificate was required for access to schools and health facilities. In addition, communities were motivated by peer pressure because everyone else in their communities was vaccinated, and by indirect pressure from HCWs telling community members to get vaccinated. Overall, reluctance to vaccinate against YF was justified by fear of side effects and mistrust of the political system. However, we only interviewed people who had been vaccinated. Therefore, we cannot provide insight into the specific reasons why some individuals refuse the vaccination. Our findings only reflect the perceptions of those who were vaccinated and participated in the study. In addition, we also interviewed elderly people about a mass vaccination campaign that had taken place in northern Uganda seven years earlier. Therefore, they may not have remembered it in detail.

This study demonstrates that people relied on community sources for information. Among our participants, access to information was unequal and decreased from urban to rural areas. This is consistent with literacy rates in Uganda [28]. Although Uganda's literacy rate has improved to 76.53% in 2018, a 6.33% increase from 2012 [37], it is still unevenly distributed across districts due to unequal access to education [28]. Furthermore, literacy rates are lowest among the elderly (60+ years) (38%), women (66–70%), and subsistence farmers (77%) [28]. In our sample, 23% of the 43 participants in the in-depth interviews had no education—all elderly women working as subsistence farmers in rural areas. Low literacy rates result in unequal access to information and may lead to vaccine refusal [38]. Low levels of education may lead to a lack of understanding of information about the vaccine and its benefits. In addition, people may be more susceptible to rumors and misinformation because they rely on other community members for information and cannot educate themselves. During the COVID-19 pandemic, Ahiakpa et al. described that young Africans with higher levels of education, such as university degrees, had the highest level of awareness through social media campaigns, local TV/radio, and newspapers, and less through community mobilization and religious gatherings [39]. This may be one of the reasons rural populations, which are at high risk of YF outbreaks, are not adequately reached by current awareness strategies.

Thus, other strategies for health education may be necessary. For example, Abiodun et al. described how they raised awareness of cervical cancer among rural women in Nigeria by showing a 25-minute health education film in the local language with a lecture on cervical cancer [40]. This was followed by a health education session with 50 women, during which they

were encouraged to answer questions. Afterward, they were given an information booklet to take home [40]. The strategy of showing a health education film was used at some IDP camps during the YF outbreak in 2010 in northern Uganda. To reach more people, such short educational films could be shown at local markets, communal areas, religious gatherings, and health centers. In addition, informational booklets in the form of comics could be distributed. These measures could improve access to information for illiterate people. For example, the impact of health education was demonstrated during the 2014 Ebola outbreak in Sierra Leone when health education in villages helped to identify suspected cases more quickly and interrupt further transmission [41]. Thus, increasing awareness in rural communities is effective, and community awareness should be raised directly through health education.

In addition, our results showed that the YF information that reached affected communities differed from the information disseminated by medical experts. YF mass vaccination was based on a voluntary approach. However, our participants perceived it as mandatory. Participants agreed to be vaccinated because HCWs told them they would die of YF if they were not vaccinated. Participants complied with this request without question. This may indicate that HCWs are trusted as experts, but it also reflects a power imbalance that may backfire in future campaigns. This aligns with Holt et al., who emphasize patient empowerment and two-way communication in immunization campaigns [42]. Although HCWs are the most trusted advisors and influencers of vaccination decisions, their ability to address vaccine issues may be limited due to increased workload and limited resources, such as inadequate information and training [43].

We found a hierarchy of knowledge between medical experts and the general population in all districts. This could be because the participants have unequal access to education [28]. Participants explained that they trusted health experts because they were more educated. In addition, a power dynamic in the public sector directly influenced what the general population believed and did not believe. Furthermore, it has been shown that power dynamics among health professionals can repress groups such as non-clinical experts and VHTs who better understand barriers to the uptake of health services [44,45]. Our results showed that because of this steep hierarchy in the public sector, several participants handed over their decision-making to the health experts. However, power dynamics between the government in central Uganda and other tribes in the north [46], as well as the political opposition in the west [47], increased distrust of the government's mass vaccination program.

According to our results, some of our study participants did not understand the purpose of YF vaccination and were unsure whether the disease was being prevented or treated. Thus, a lack of understanding of the vaccine may prevent informed decision-making and negatively affect vaccine uptake [48]. For example, although YF vaccines are available free of charge in Nigeria, low vaccine uptake has led to frequent outbreaks, with communities opting for traditional medicine to treat YF rather than vaccination as a preventive measure [49]. Thus, people need to be adequately informed about the safety and purpose of vaccines to make an informed decision about vaccination. Health messages should address the information needs of the target population, use language that the target group understands, use credible sources, and be brief without overemphasizing the health benefits of vaccination [50].

Furthermore, our study showed that residents of unaffected districts rushed to get vaccinated. They were motivated by access to an expensive vaccine required for international travel. This may have targeted more educated people with higher socioeconomic status and means for international travel rather than communities vulnerable to YF infection. As our results have shown, some communities believed that they were being targeted for death by being given the YF virus instead of the vaccine. This perception of being harmed by vaccination programs is consistent with the findings of Braka et al., where caretakers in Uganda held

misconceptions about childhood vaccination programs, that the vaccine would cause infertility, that children would contract HIV, and that they would be killed because they were African [51]. Furthermore, in Zambia, community members and HCWs perceived that Western countries brought COVID-19 vaccines to eliminate the African population and that certain vaccines, which Western nations had rejected, were given to Africans [52]. It is essential to understand these vaccine rumors because they reflect a power imbalance [53].

Our findings showed that the YF mass vaccination campaign was strongly linked to local politics. The Ministry of Health carried out the campaign, which was perceived as an activity directly implemented by President Museveni and his ruling party. YF mass immunization was politicized because yellow and blue are associated with political parties in Uganda—yellow represents the ruling party, and blue represents the opposition. Opposition community members did not want to be associated with the ruling party by having a yellow vaccination card. They refused the YF vaccine for political reasons. In addition, some districts conducted mass immunization campaigns during or after the presidential election. The impact of local politics on vaccination campaigns is important and should not be underestimated. For instance, during the COVID-19 pandemic, some citizens in Zambia believed that COVID-19 vaccines were introduced for political reasons because 2021 was a presidential election year. Zambian politicians were perceived to have a contract with Western countries to administer the COVID-19 vaccine in exchange for money to fund their election campaigns [52].

Vaccine hesitancy has been viewed primarily as a problem of misinformation, but trust is a key factor in overcoming vaccine hesitancy [54]. Trust is related to past experience, with marginalized groups already more susceptible to rumors and hesitancy (40). According to Pertwee et al., conspiracy theories about COVID-19 and vaccines are not simply false beliefs but expressions of widespread fears and anxieties in times of acute social insecurity, lack of public trust in institutions, and distrust of political elites and medical experts [54]. Thus, vaccine hesitancy should be seen in the context of social crisis and trust. Therefore, strengthening vaccine programs requires trust and relationship building, such as training HCWs from local communities [54]. In Nigeria and Pakistan, the politicization of vaccine campaigns led to the failure of the global polio eradication initiative due to rumors spread by religious and political leaders. Muslim fundamentalists perceived the polio vaccine as a Western plot to sterilize the Muslim population [55]. In Pakistan, the politicization of vaccination campaigns culminated when the CIA infiltrated a hepatitis B vaccination campaign to locate and kill Osama Bin Laden. This ruse and abuse of power had a lasting negative effect on public confidence in health services, and demand for vaccination in the country declined [56].

Vaccine rumors spread by religious and political leaders are of concern because our results showed that study participants of low socioeconomic status in remote areas relied heavily on information from trusted community members, often religious and political leaders. Moreover, the politicization of vaccine campaigns can lead to large-scale changes at the global level. For example, local politics in Nigeria and Pakistan led to a decline in polio immunization campaigns [55]. This immunity gap increased vaccine-derived polio cases in other countries with low immunity [57,58] and the resurgence of polio in eight previously polio-free countries in Africa [59]. Thus, the failure of politically motivated vaccination campaigns in one country can have a snowball effect with broader implications for global disease control efforts. Although immunization campaigns can never be seen as entirely detached from political systems and power dynamics, they should not be deliberately abused for political reasons, as was the case in Pakistan.

Previous studies have shown that a mismatch between frequent, well-funded polio campaigns and underfunded health care in government health facilities created distrust among parents who feared that a vaccination campaign might not be what it seemed [55]. Our results

showed that two different immunization campaigns could interfere with each other if they were conducted shortly after each other. As a result, one community was suspicious when HCWs returned for another immunization campaign (YF) after the mass polio immunization for children had just been completed. In addition, adults were being targeted for mass vaccination for the first time in the region. This led to the belief that the purpose of the new YF vaccination campaign was to kill all community members.

Our results showed that the proximity of different vaccination campaigns, new target populations, and insufficient information reaching remote villages increased vaccine rumors within the community. This is consistent with previous findings on campaign interference described by Feldman-Savelsberg et al. in Cameroon [60]. There, a preceding family planning campaign and increasing political unrest were shown to led to rumors that the administration of a subsequent tetanus vaccine would sterilize young women. In addition, only young women of childbearing age were targeted for tetanus vaccination. The tetanus vaccine was falsely believed to cause infertility among teenage girls, leading to disastrous consequences such as unwanted pregnancies and botched abortions. Teenagers tested their fertility to confirm the rumors, increasing teen pregnancies. The termination of unwanted pregnancies with botched abortions further exacerbated the issue, ultimately leading to increased infertility among vaccinated teenage girls. Overall, this had a long-lasting effect on subsequent immunization campaigns, with setbacks in vaccine uptake and diminished confidence in vaccination programs [60].

## Study limitations

YF outbreaks and mass vaccination are dramatic events. However, it is important to recognize that recall bias is a significant limitation, especially at study site 1, where data were collected seven years after the outbreak. Additionally, only elderly individuals were interviewed in this region, which may have impaired the reliability of the data as they may not have remembered in detail. The study only included individuals who were vaccinated against YF during reactive mass vaccination campaigns. Therefore, these participants may have a generally more positive attitude towards vaccination. The results solely reflect the perceptions of those who participated in the study. It is important to note that the study cannot provide any insight into the reasons for refusing vaccination since only vaccinated individuals were included. The power dynamics created by an outsider, a white female researcher, may have shaped the understanding of the research topics. In addition, gender dynamics within the society may have influenced the research. Our findings cannot be generalized to other countries because we experienced power relations specific to Uganda, such as the north-south divide, local politics, lack of social competence in the public sector (the inability of the health sector to respond to the needs of the population), and Bantu leadership in the north, where another tribe lives. Thus, data collection in districts with power dynamics, such as the opposing political party and the north-south divide, may have influenced our findings on the politicization of YF vaccination campaigns. However, the findings on top-down communication by HCWs may apply to other regions and countries. Another limitation is that we did not conduct member-checking. This would have been almost impossible because we conducted our data collection in remote rural villages where many participants were illiterate and without access to electricity. Thus, we would not have been able to reach them by phone or email. We would have had to meet them in person, but when it's not the rainy season, people are scattered, working in their fields, and it's challenging to meet.

## Conclusion and recommendation

People in remote areas affected by YF outbreaks rely on community sources of information. Therefore, we recommend improving health education and communication through respected

community members. In particular, a participatory approach should be used for vaccination campaigns. For example, medical experts and respected leaders should be involved in planning and implementing campaigns in their districts. As our results showed, the involvement of local experts and leaders in western Uganda was critical to the campaign's success. Thus, people who are respected and trusted by the community should be involved in the campaigns. However, despite extensive awareness campaigns during the reactive emergency YF mass vaccination, HCWs and affected communities were not adequately informed. In Uganda, HCWs are often deployed to other regions where they are outsiders and unfamiliar with the culture and customs. This can create a communication gap between HCWs and communities. We recommend that HCWs be trained in social skills and how to communicate in a positive way to increase vaccine uptake [42].

In addition, YF vaccination was politicized and led to refusals because of the association of yellow cards with the ruling political party. The political systems and power relations should be considered when implementing vaccination campaigns. Thus, we recommend that awareness campaigns focus on understanding the importance of vaccination and that the vaccination card is yellow according to international standards rather than on having a YF certificate for travel. Understanding the health messages being communicated is essential to make an informed decision about whether or not to be vaccinated. Therefore, health messages should use language understood by the people intended to reach [42]. Linguistically, it should also be understandable to health workers at all levels. In particular, it should be taken into consideration that HCWs often have only two years of training.

Moreover, it is essential to understand local perceptions of vaccination. For example, when routine immunization was reintroduced in Liberia after the Ebola outbreak, some of the key barriers found in communities with low vaccine uptake included suspicion that the measles vaccine was the experimental Ebola vaccine and a way to infect children with Ebola [61]. Thus, understanding these barriers and driving factors can improve access to immunization and increase vaccine uptake. Thereby, it can also raise equity in the health system. When two conflicting campaigns, such as a vaccination campaign and an election campaign, are conducted simultaneously or in quick succession, it is important to provide clear and detailed information to explain the reasons behind each campaign. Linkages between campaigns should be avoided. The aim should be for each campaign to be seen as separate and performed on its own merits. In addition, when a new target group is selected for vaccination, communities must be given adequate and detailed information to make an informed decision. We also recommend that mass vaccination campaigns be monitored. As risk perception and vaccine confidence can influence an individual's willingness to be vaccinated [61,62], it is therefore important to know what information is being received by the population so that health education can be adapted accordingly. When mass vaccination campaigns are monitored, they can be evaluated, and this information can be used to improve future campaigns.

## Supporting information

**S1 File. Interview questionnaires in English, Luganda, Acholi, and Rukiga.**
(DOCX)

**S2 File. Interview guide semi-structured interviews.**
(DOCX)

**S3 File. Interview guide expert interviews.**
(DOCX)

**S4 File. Interview guide FGDs with pregnant women and elderly.**
(DOCX)

## Acknowledgments

We want to thank the District Health Officers and Local Chairpersons of Kitgum, Lamwo, Pader, Masaka, Kalangala, and Rukungiri Districts for allowing access to their districts and communities. We would also like to thank all other members of the District Health Teams and Village Health Teams for their support during data collection. Furthermore, we would like to thank Patricia Apoko and Denis Okello for their assistance in the field.

## Author Contributions

**Conceptualization:** Lena Huebl, Aloysious Nnyombi, Eddy Walakira, Ruth Kutalek.

**Formal analysis:** Lena Huebl, Ruth Kutalek.

**Investigation:** Lena Huebl.

**Methodology:** Lena Huebl, Aban Kihumuro, Denis Lukwago, Ruth Kutalek.

**Supervision:** Lena Huebl, Eddy Walakira, Ruth Kutalek.

**Writing – original draft:** Lena Huebl.

**Writing – review & editing:** Lena Huebl, Aloysious Nnyombi, Aban Kihumuro, Denis Lukwago, Eddy Walakira, Ruth Kutalek.

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
