## [Decision Letter · Decision Letter 0]

2 Jun 2023

Dear Dr. med. univ. Huebl,

Thank you very much for submitting your manuscript "Perceptions of yellow fever emergency mass vaccinations in Uganda: A qualitative study" for consideration at PLOS Neglected Tropical Diseases. As with all papers reviewed by the journal, your manuscript was reviewed by members of the editorial board and by several independent reviewers. In light of the reviews (below this email), we would like to invite the resubmission of a significantly-revised version that takes into account the reviewers' comments. 

We cannot make any decision about publication until we have seen the revised manuscript and your response to the reviewers' comments. Your revised manuscript is also likely to be sent to reviewers for further evaluation.

Sincerely,

William B Messer

Academic Editor

Abdallah Samy

Section Editor

Reviewer's Responses to Questions

**Key Review Criteria Required for Acceptance?**

**Methods**

-Are the objectives of the study clearly articulated with a clear testable hypothesis stated?

-Is the study design appropriate to address the stated objectives?

-Is the population clearly described and appropriate for the hypothesis being tested?

-Is the sample size sufficient to ensure adequate power to address the hypothesis being tested?

-Were correct statistical analysis used to support conclusions?

-Are there concerns about ethical or regulatory requirements being met?

Reviewer #1: This is a qualitative study exploring a highly critical topic, applying mixed methods for data collection and analyses. The methods section is short and would benefit from more detailed descriptions in several areas:

It is not clear how participants were selected for either of the data collection methods. The study aims to describe vaccine implementation, but has not included any non-vaccinated in the study population. Suggest to describe clearly recruitment, and why, if a purposive sample typical for qual studies, no non-vaccinated were included. 

It is not clear how analysis has been done. Inductive content as well as grounded theory are mentioned. The results section is however not including any emerging themes, categories or similar, but is rather a descriptive account of “s/he said..” which depicts it as closer to a presentation of manifest content rather than at analytical level looking at hidden mechanisms. This is specifically evident eg for the power perspectives discussed related to political context, which is relying on direct citations from key informants rather than applying a more analytical approach. It seems the various methods and informants included could open up for triangulation and more analytical approaches and my recommendation would be to explore of possible to provide a more elaborated analysis based on that.

It is also very evident from the lack of non-vaccinated participants, that the interpretations on causes for vaccine refusal or reluctance are based on opinions of the vaccinated participants, and what they think are causes and reasons, this makes the analysis of vaccine barriers somewhat less interesting. 

I may have missed but couldn’t find a section on robustness or reflectivity, neither anything on how and in which language the constant comparisons were done. I note the first and last/main authors are affiliated w European institutions, and a reflective section on how the team has worked together and the outsider/insider roles in analysis would be beneficial.

Reviewer #2: Overall, the methods section is acceptable in that it is made explicitly clear where the research was conducted, with some additional information helping contextualize the socio-political context of the sites. It’s also clear that appropriate ethical approval was gained. However, the specific type of research design adopted should be outlined more clearly at the start of the methods section, especially in terms of how such an approach helped address the overarching research question. Also, more reflection is needed on the relationship between the researchers and the researched – i.e., possible power dynamics in this dynamic need to be reflected upon. Tying in with the latter point, can you state whether any participants left whilst the research was being conducted. It’s also worth defining more clearly the specific sampling approach you adopted for this project. Some more specific points include:

Line 172 – Can you be more specific relating to who did what during the analysis process (using author initials). 

Line 174 – Please provide one example from your coding which illustrates this analytical process.

Line 185 – I don’t like the term ‘guarantee anonymity’ because can this really be the case in any research project? Try rephrasing this using softer terms.

Line 188 – Can you be a little clearer about what exactly participants received (i.e, a gift voucher worth X amount).

**Results**

-Does the analysis presented match the analysis plan?

-Are the results clearly and completely presented?

-Are the figures (Tables, Images) of sufficient quality for clarity?

Reviewer #1: The results section is per above mostly purely descriptive including citations. No analytical categories are presented. It is a little difficult to assess the results outside the manifest in tent of the included citations. Results could benefit from sharing the work mentioned done with grounded theory and identification of emerging categories. For the power analyses it would also benefit from understanding better what are direct experiences, norms and thoughts reported, what are those where participants tell “what others think” and what are key informants reflecting from professional perspectives given power of their roles.

Reviewer #2: The results section outlines an excellent volume of data that is directly related to the research question. Also, it is good that characteristics of respondents included in the study are clearly outlined at the start of this section. However, this section needs more structural work in terms of flowing between themes in helping the reader process what is being said. A thematic table (which includes thematic examples) at the start of this section could help provide such structure. Also, I was a little bit confused with the theme ‘Perception of YF mass immunization’, as it thematically mirrors the theme ‘Motivation for YF vaccination’. For instance, see primary data cited on line 482 - in terms of meaning can this not relate to motivation against vaccination? Could these themes be integrated somehow? Some more specific points include: 

Table 2 (Participants of focus group discussions) – Pregnant women median age cell should be 30.5, not 30,5. 

Table 3 (Participants of expert interviews) – Relating to occupation, have you spelled out the acronym DHO before using it here?

Line 211 – The acronym Village Health Teams has already been spelled out on line 158.

Line 231 - The acronym Internally displaced people (IDP) has already been spelled out on line 80. 

Line 260 – Not sure what you mean by gas here? Do you mean fuel, as in petrol or diesel?

**Conclusions**

-Are the conclusions supported by the data presented?

-Are the limitations of analysis clearly described?

-Do the authors discuss how these data can be helpful to advance our understanding of the topic under study?

-Is public health relevance addressed?

Reviewer #1: The conclusions are somewhat supported, but more underscore the experiences of some participants. The experiences of others stating vaccine willingness “because they are told to do it” could perhaps be elaborated more. Further would find it interesting if conclusions would discuss suggestions beyond community mobilisation, especially related to the political trust- the opposition+s reluctance to vaccinate comes fw strongly and I would be curious to understand what insightful key informants would suggest related to that.

Reviewer #2: The discussion and conclusion sections are appropriate in terms of reflecting on and answering the set research question. Broader implications of findings are also discussed within the context of relevant literature. Further, authors should be particularly commended for their interesting take on the politicization of vaccine campaigns relating to YF immunization, which I feel is very insightful. However, a major issue within this section is the lack of reflection on study limitations. More has to be said regarding key methodological considerations. For instance, how transferable are the findings you produced? Was any member checking conducted during data analysis and write up? Is there the possibility that sample characteristics may have shaped your findings in a certain way? More reflection must be provided on these issues. Also, can the discussion section be split up into different paragraphs to help provide the reader with more structure? Because the section is one ‘whole’, sometimes it gets a little confusing for the reader. For example, it would make sense to start a new paragraph when you change the topic of discussion, as on line 607. 

Some more specific points include: 

Line 527 – Can you please cite the direct source of this statistic ... which I think going by the link you provided here is the world bank.

Line 540 – Can you provide a citation or example of how such health education interventions may look like? Do any examples exist from other VHF outbreaks? 

Line 641 – Can you add a sentence or two here clarifying what exactly you mean ... local perceptions rooted to specific community needs ... can you support this point with a practical example or citation?

**Editorial and Data Presentation Modifications?**

Reviewer #1: Interview guide/questionnaire should be appended.

Use sub-headings in methods to cover critical parts.

Reviewer #2: The overall grammatical/sentence structure of the manuscript is ok, but before publication I think it will have to be much more polished. This point must be stressed to the authors, as it shouldn’t take long to go through the manuscript again in making sure the quality of writing is up to the journal’s standard.

**Summary and General Comments**

Reviewer #1: this is a critical area of study and interesting indeed to see results closely linked to mass vacc campaigns. The paper will be able to contribute important knowledge after major revisions per above. The suggested major recommendation could be problematised, ie indeed empowered patients asking questions and seeking information is an ideal situation, and could also be illuminated versus the public health perspective of disease control, and asking for “unlimited trust” in public measures, eg Covid-19, without vaccination freedom of mobility was challenged etc.

Reviewer #2: Overall, this is a great contribution to the VHF literature base more broadly, and YF evidence base more specifically. The use of qualitative methods here produced interesting results, and the comparative study design allowed for the exploration of such results between two different sites. Findings were then interpreted with relevant literature, and implications for practice were sufficiently reflected upon. Although there are areas that need improved (such as overall sentence structure and methodological reflection), this work will be of interest to practitioners and policy makers who specialize in infectious disease, potentially helping inform future vaccination campaigns. For this reason, after the recommended changes have been made, I recommend this manuscript be published.

PLOS authors have the option to publish the peer review history of their article (what does this mean?). If published, this will include your full peer review and any attached files.

Reviewer #1: No

Reviewer #2: Yes: Anthony John Bell
---

## [Decision Letter · Decision Letter 1]

27 Jan 2024

Dear Dr. med. univ. Huebl,

Thank you very much for submitting your manuscript "Perceptions of yellow fever emergency mass vaccinations in Uganda: A qualitative study" for consideration at PLOS Neglected Tropical Diseases. As with all papers reviewed by the journal, your manuscript was reviewed by members of the editorial board and by several independent reviewers. The reviewers appreciated the attention to an important topic. Based on the reviews, we are likely to accept this manuscript for publication, providing that you modify the manuscript according to the review recommendations. 

Sincerely,

Willian B. Messer

Academic Editor 

Abdallah Samy

Section Editor

Your response to reviewer feedback was both comprehensive and detailed. The manuscript reads well and we do not have additional content changes to request.

Reviewer's Responses to Questions

**Key Review Criteria Required for Acceptance?**

**Methods**

-Are the objectives of the study clearly articulated with a clear testable hypothesis stated?

-Is the study design appropriate to address the stated objectives?

-Is the population clearly described and appropriate for the hypothesis being tested?

-Is the sample size sufficient to ensure adequate power to address the hypothesis being tested?

-Were correct statistical analysis used to support conclusions?

-Are there concerns about ethical or regulatory requirements being met?

Reviewer #3: Lines 137-138: “We only included people who were vaccinated during YF mass vaccination campaigns because the aim was to better understand the perception of YF vaccination” – this doesn’t make sense. It would also have been useful to understand the perceptions of those who did NOT accept the vaccination for whatever reasons. This should be recognised as a limitation of this study in the discussion. It is mentioned too briefly in the “limitations” section at the moment. Using this method, it will not be possible to gain a deep understanding of reasons for vaccine refusal.

Lines 139-140: “People who were vaccinated received a YF card. This made it easy to verify who had been vaccinated and at what time.” – had everyone kept their YF card carefully? Did nobody lose it?? This sounds improbable to me. So if you are selecting people with a card, you will have a selection bias in favour of individuals who are very well organised and take their health seriously. This should also be recognised as a limitation in the discussion. 

Lines 141-142: “It would have been difficult to detect unvaccinated individuals because our study was conducted several months after the YF mass vaccination, and in northern Uganda even years later.” - this does not make sense to me. If you are saying that people can’t remember whether or not they had been vaccinated with YF, then how do you think they will be able to answer detailed questions about their perceptions of YF vaccine several years later? I note that the YF vaccination campaign in the North was in 2010 and the study was conducted in 2017. How can you be confident that the recollections of people will be accurate, 7 years later? I think the authors need to be more self-critical about the reliability of this information. 

E.g. in the results, line 369-370: “When we were being vaccinated, there wasn’t adequate information about yellow fever. They did not give us any information apart from just the call to people to come and get vaccinated”

- Do you think that this information is reliable, 7 years after the event, in a group of elderly men (whose memory is probably less than perfect)? 

“The focus was not on vaccine refusal, but rather on understanding perceptions of YF vaccination to inform future YF mass immunization and thereby optimize vaccination campaigns.” – to me this doesn’t make sense. Vaccine refusal is part of the perceptions of vaccination, and is key for informing future campaigns. Surely one of the main aims of future campaigns is to reach the maximum number of people? So how can this be achieved without addressing reasons for vaccine refusal?

Line 161: “were purposively selected from six affected districts” – what was the purposive sampling frame? I can see in table 2 that the median age was 70 which is quite old. Did you have adequate representation of younger age groups too? 

Table 3 suggests that elderly men were only recruited to an FGD in the north, and pregnant women only in the south. Is it possible that pregnant women in the north maya have different views? And elderly men in the south? The languages and cultures are very different. 

Lines 217-218: “The LH transcribed all semi-structured interviews and expert interviews.” – Is LH the first author? (in which case, remove “the”). Is she fluent in Acholi, Luganda and Rukiga? Otherwise how could she transcribe all the interviews?

Data analysis: it seems that this was all done by the first author. Were Ugandan colleagues involved in this at all, in any way? 

Table 1 is very good but it would be better to place it in the results section rather than the methods. In fact I would suggest it replaces Fig 3 because they both have the same information, and the table is clearer to me than the figure. 

The section on robustness and reflexivity is good. However, it should also consider the extent to which participants were able to recall events 7 years earlier (in the case of Northern Uganda). It should also consider the positionality of Ugandan co-investigators.

**Results**

-Does the analysis presented match the analysis plan?

-Are the results clearly and completely presented?

-Are the figures (Tables, Images) of sufficient quality for clarity?

Reviewer #3: Lines 345-6: “In addition, posters were displayed at health facilities and vaccination sites in the local language, Acholi, at study site 1 and in English at study site 2.”

- Did you ask participants what they understood from these posters, especially the ones in English? Did they interpret them correctly?

Lines 369 – 381 – you take at face value the quotes and assume it is true that less information was given in the north than in the south. However, an alternative explanation could be that the elderly men in the north had forgotten the details of what happened 7 years ago, whereas the younger women in the south had a much clearer recollection of what had happened the previous year. 

In fact, the recollection of the elderly men is clearly incorrect, because the elderly women (also in the north) remembered watching an informative film about YF (previous quote).

Also – good information was not universal in the south – as demonstrated by the quote in lines 775-779

Line 499 and following– from which study site is the female health worker with 2 years’ experience? Would be good to state this for all the medical experts.

Lines 581-583 – this quote is strange. It is from an 84 year old man referring to his children being at boarding school. Does he mean his grandchildren or even great-grandchildren? His children must be adults!

Line 738 – there is a mistake here. It should be NRM not NRF, and it stands for National Resistance Movement - https://www.nrm.ug/

Line 740 – should this be RDC not FDC? If it is FDC, what does this stand for?

**Conclusions**

-Are the conclusions supported by the data presented?

-Are the limitations of analysis clearly described?

-Do the authors discuss how these data can be helpful to advance our understanding of the topic under study?

-Is public health relevance addressed?

Reviewer #3: Lines 949 – 951: “Vaccination campaigns can therefore never be considered independent of political systems and power relations. Thus, we recommend that awareness campaigns focus on understanding the importance of vaccination rather than on having a YF certificate for travel”

- The first sentence seems to be an over-generalisation – this seems a very bold statement, and I’m not sure it can be justified on the basis of this small study in a very specific context. For example I doubt that childhood vaccinations are seen as part of the political system in most countries. In the example of COVID vaccines, while there were all sorts of conspiracy theories, not everyone believed in them. 

- Why not include awareness in the campaigns of the real reason why the card is yellow, and that this is an international standard (not only in Uganda)? 

Lines 957 – 8: “For example, in Liberia after the Ebola outbreak, when routine immunization was reintroduced.” – this is not a complete sentence.

Lines 962 – 4: “We also recommend that two conflicting campaigns, whether another immunization campaign or an election campaign, should not be conducted at the same time or in close proximity.” – that’s all very well in theory, but what do you do if you get a YF outbreak just after you’ve finished another vaccination campaign, and it happens to also be an election year? One cannot plan for outbreaks in advance, and surely it is still better to conduct the mass immunisation than not to do it.

So – rather than saying not to do this, perhaps this should just be a recommendation for better information to explain the reasons.

**Editorial and Data Presentation Modifications?**

Reviewer #3: (No Response)

**Summary and General Comments**

Reviewer #3: I note that this version already incorporates revisions in response to previous reviews. I have not reviewed this manuscript before, so I apologise if some of these suggestions were not made previously. 

Overall this article is well-written and deserves to be published. 

The major issues are 

1. Sampling bias – only vaccinated people with a card were interviewed

2. Recall bias – participants in the North were interviewed 7 years after the vaccination! And they were mainly elderly people with less than perfect memories. This should be recognised as a major limitation.

3. Reflexivity – this section focusses only on the first author. The positionality of Ugandan co-investigators and colleagues should also be mentioned. 

It is now too late to address the bias in the design but it should be discussed more extensively in the discussion, and the authors should recognise that this may influence their interpretation of any differences between the North and South – which may be partly due to greater recall bias in the North. 

Specific comments

Abstract:

Methodology – add a sentence about the analytical method

Conclusion – should be “We therefore recommend…” – (remove commas)

Intro

Line 94-95: “the YF vaccine is planned to be introduced into the routine immunization program in mid- 2022” – please update to state whether or not this has happened. 2022 is now in the past not in the future!! 

Discussion

Line 881 – you suggest a film but in fact quotes above say that this was already done. Perhaps the discussion should consider how such films could be shown more widely to reach more people. 

Line 813 – you mention a booklet to take home – but is this likely to help in areas with such high levels of illiteracy as you have just stated above?

Lines 821-822 “Participants complied with this request without question. This may indicate that HCWs are trusted as experts, but it also reflects a power imbalance that may backfire in future campaigns”- this is because you only interviewed people who had been vaccinated! So what it really reflects is sampling bias in your sample!

Lines 856 – 858: “Furthermore, in Zambia, community members and HCWs perceived that Western countries brought COVID-19 vaccines to eliminate the African population and that certain vaccines were given to Africans who were rejected by Western countries.” – is it the Africans who were rejected by Western countries, or the vaccines?

Lines 911-912: “This had disastrous consequences with unwanted teenage pregnancies due to fertility testing and increased infertility due to botched abortions” – this doesn’t make sense. Why would fertility testing cause unwanted teenage pregnancies?

Line 924: “social incompetence in the public sector” – what do you mean by this?

PLOS authors have the option to publish the peer review history of their article (what does this mean?). If published, this will include your full peer review and any attached files.

Reviewer #3: Yes: Merlin Willcox

Figure Files:

Data Requirements:

Reproducibility:

To enhance the reproducibility of your results, we recommend that you deposit your laboratory protocols in protocols.io, where a protocol can be assigned its own ident

---

## [Editor Report · Decision Letter 2]

29 Apr 2024

Dear Dr. med. univ. Huebl,

We are pleased to inform you that your manuscript 'Perceptions of Yellow Fever Emergency Mass Vaccinations Among Vulnerable Groups in Uganda: A Qualitative Study' has been provisionally accepted for publication in PLOS Neglected Tropical Diseases.

Best regards,

William B Messer

Academic Editor

Paul Brindley 

EIC

---

## [Editor Report · Acceptance letter]

7 May 2024

Dear Dr. med. univ. Huebl,

We are delighted to inform you that your manuscript, "Perceptions of Yellow Fever Emergency Mass Vaccinations Among Vulnerable Groups in Uganda: A Qualitative Study," has been formally accepted for publication in PLOS Neglected Tropical Diseases.

Best regards,

Shaden Kamhawi

co-Editor-in-Chief

Paul Brindley

co-Editor-in-Chief
